# On Thermal Infrared Remote Sensing of Plastic Pollution in Natural Waters

**Lonneke Goddijn-Murphy \*** and **Benjamin Williamson**

Environmental Research Institute, North Highland College, University of the Highlands and Islands,
Thurso KW14 7EE, UK; benjamin.williamson@uhi.ac.uk
**\*** Correspondence: lonneke.goddijn-murphy@uhi.ac.uk; Tel.: +44-(0)18-4788-9664

**Abstract:** Plastic pollution in the world's natural waters is of growing concern and currently receiving significant attention. However, remote sensing of marine plastic litter is still in the developmental stage. Most progress has been made in spectral remote sensing using visible to short-wave infrared wavelengths where optical physics applies. Thermal infrared (TIR) sensing could potentially monitor plastic water pollution but has not been studied in detail. We applied radiative transfer theory to predict TIR sensitivity to changes in the surface fraction of water covered by plastic litter and found that the temperature difference between the water surface and the surroundings controls the TIR signal. Hence, we mapped this difference for various months and times of the day using global SST (sea surface temperature) and *t2m* (temperature at 2 m height) hourly estimates from the European Centre for Medium-Range Weather Forecasts (ECMWF), ERA5. The maps show how SST-*t2m* difference varied, altering the anticipated effectivity of TIR floating plastic litter remote sensing. We selected several locations of interest to predict the effectivity of TIR sensing of the plastic surface fraction. TIR remote sensing has promising potential and is expected to be more effective in areas with a high air–sea temperature difference.

**Keywords:** plastic litter; thermal infrared; natural waters; pollution

## 1. Introduction

Plastic pollution in the world's oceans rivers and lakes is of growing concern and currently receiving significant attention, but there are still many unknowns. Every year 4.8 to 12.7 million metric tons of plastic is estimated to enter the oceans from land [1] of which only about 1% is found floating at sea [2]. Plastic can last a very long time in the environment, and it is not clear where all the plastic goes. Known removal processes of marine plastic litter are sinking to the ocean floor, stranding, chemical degradation under the influence of ultraviolet light from the sun, fragmentation by breaking waves, and ingestion by animals. Within the centres of the gyres where plastic debris accumulates, existing ocean circulation models agree reasonably well, but in other areas, they can differ by more than a factor of 100 [3]. Remote sensing is an effective tool to monitor the ocean surface on local to global scales and has many applications, but remote sensing of marine plastic litter is still in the developmental stage. Most progress has been made in the field of spectral remote sensing in visible (VIS), near-infrared (NIR) and short-wave infrared (SWIR) spectra where optical physics applies. Over these wavelengths, spectral light reflectance measurements of floating plastic litter have been made in situ [4], from unmanned aerial vehicles (UAV) [5], airplanes [6] and very recently, the Sentinel-2 satellite [5]. It has become clear that spectral remote sensing would be improved by using complementary measurements using different sensing technologies [7]. Thermal sensing operating in the thermal infrared (TIR) spectrum has been proposed as one of those technologies. For example, materials that are transparent for optical wavelengths can be opaque for thermal wavelengths and

vice versa. In addition, TIR sensing is (unlike optical spectroradiometry) completely passive with no external illumination, such as sunlight, required since the sensor records the energy directly from the object. TIR sensing can, therefore, perform during both day and night. At the longer end of the TIR spectrum, it does not need clear weather, and solar rays are diffusely reflected from the water surface, which reduces the sensitivity to solar glint.

The potential of TIR as a new remote sensing technology for floating plastic litter is based on the different emissivity values of water and of plastic. Emissivity is the ratio of the energy radiated from a surface, and that radiated from a black body. The emissivity of water is very high, near one, and of plastics, it is generally lower. The warmer an object, the higher the emitted radiance, and this radiance increases with increasing emissivity. So, for water and plastic at the same temperature, the emitted radiance will be stronger for water. However, the water and plastic surfaces not only emit their own thermal radiance, they also reflect thermal radiance from their surroundings. For an object of lower emissivity, the reflectivity is higher, which implies that plastic is generally a stronger reflector than water. Water will, therefore, more closely indicate the actual temperature of the water while plastic, the temperature of the surroundings. In conclusion, the thermal infrared signal of plastic is expected to improve with a lower value of plastic emissivity, enhanced by an increasing temperature difference between the air and sea. This dependence on air and sea temperatures complicates interpretations of the TIR signal, but it could also be used in our favour, for example, by using a difference between day and night measurements to retrieve plastic presence. TIR cameras have been used to image marine plastic litter from the air [5], but a thermal radiative transfer model or a remote sensing algorithm that quantifies surface plastic does not yet exist.

This paper describes the theory that can explain the physics of TIR remote sensing of plastic litter floating on a water surface and help interpret thermal images captured by TIR cameras. Two types of TIR cameras operate in different atmospheric windows of the TIR spectrum: medium-wave infrared (MWIR) 3–5 μm, and long-wave infrared (LWIR) 8–14 μm [8]. Each has its own advantages and disadvantages depending on the application. MWIR detectors are generally sensitive and fast but are heavier and more expensive, mainly due to cooling requirements. LWIR detectors are noisier and slower, but cost less, are lighter and more robust [9]. MWIR works better under clear skies, while LWIR works better in fog and dust conditions. Another advantage of LWIR is that sun glint is nearly negligible [9]. Remote sensing of marine plastic litter based on TIR could be applied to narrow-band measurements, such as from space, that are already available. These could supplement observations in the VIS-NIR-SWIR spectrum for which remote sensing of marine plastic litter is further advanced and other Earth observation data.

## 2. Materials and Methods

### 2.1. Radiative Transfer Theory

According to Planck's law, the spectral radiance, $L$, of a black body for temperature, $T$ [K], and wavelength, $\lambda$ [μm] increases with increasing temperature according to Equation (1).

$$L(\lambda, T) = \frac{2hc^2}{\lambda^5}\left(\frac{1}{e^{hc/\lambda k_B T} - 1}\right)10^{24} \left[W\ m^{-2}sr^{-1}\mu m^{-1}\right], \tag{1}$$

with $h$ Planck's constant ($6.626 \cdot 10^{-34}$ J s), $c$ the speed of light ($299792458$ m s$^{-1}$) and $k_B$ the Boltzmann constant ($1.3806 \cdot 10^{-23}$ J K$^{-1}$). For a spectral band with wavelengths from $\lambda_1$ to $\lambda_2$ μm we then calculate band radiance, $L_b$, as

$$L_b(T) = \int_{\lambda_1}^{\lambda_2} L(\lambda, T)d\lambda \left[W\ m^{-2}sr^{-1}\right]. \tag{2}$$

The corresponding band emittance (radiant flux emitted by a surface per unit area), $M_b(T)$ [Wm$^{-2}$], equals $\pi L_b$ for a Lambertian surface, meaning the emitted radiance is the same in all directions. This is

a common approximation in TIR remote sensing and reasonably valid for a water surface, but for other surfaces, such as grass and bare soil, less certain [10]. Tu et al. [10] found that surface smoothness enhances the Lambertian property for all surfaces, and we assume the same applies to plastic litter. Emissivity, $\varepsilon(\lambda)$, is the ratio of energy radiated from a surface at a given temperature and then radiated from a black body at the same temperature (Equation (3)).

$$\varepsilon(\lambda) = \frac{M_{b,obj}}{M_b},\tag{3}$$

For a body in thermal equilibrium with its surroundings, all radiation that is absorbed is emitted again so that $\varepsilon(\lambda)$ equals absorptivity $\alpha(\lambda)$, the fraction of radiance absorbed, (Kirchhoff's law). The conservation of energy dictates that the sum of absorptivity, reflectivity ($\rho$) and transmissivity ($\tau$), equals one ($\rho/\tau$ represents the fraction of radiance reflected/transmitted). In this current paper, we treat seawater and plastic as opaque ($\tau = 0$) and together with $\varepsilon(\lambda) = \alpha(\lambda)$ we derive Equation (4).

$$\varepsilon(\lambda) + \rho(\lambda) = 1,\tag{4}$$

Seawater can be considered opaque as water strongly absorbs thermal infrared. We assume plastic litter is also opaque in TIR, but this is not certain for all kinds. The emission of thermal radiance by transparent materials is beyond the scope of this paper, and for a review, we refer the reader to Gardon [11]. Assuming that emitted and reflected radiances of an object (the object can be water or plastic) are Lambertian, we calculated total radiance captured by a TIR sensor, $L_{b,obj}$, using

$$L_{b,obj} = \varepsilon_{obj} \cdot \tau_{atm} \cdot L_b\left(T_{obj}\right) + \left(1 - \varepsilon_{obj}\right) \cdot \tau_{atm} \cdot L_b(T_{sur}) + (1 - \tau_{atm})L_b(T_{atm}),\tag{5}$$

with $L_b(T)$ from Equations (1) and (2) [8,9]. In Equation (5), the first term represents thermal radiance emitted by an object of temperature $T_{obj}$, the second term, the thermal radiance of the surroundings of temperature, $T_{sur}$, reflected by the object, and the third term emitted radiation resulting from absorption of radiation in the atmosphere, and $\tau_{atm}$ the transmissivity through the atmosphere with temperature $T_{atm}$ between object and sensor. If $x_p$ is the fraction of the surface captured by the sensor covered with plastic and $(1 - x_p)$ the surface fraction of open water, then total TIR received by the sensor is

$$L_{b,tot} = x_p L_{b,p} + \left(1 - x_p\right)L_{b,w},\tag{6}$$

with $L_{b,p}$ and $L_{b,w}$ estimated by Equation (5) for plastic and water, respectively, and $L_b$ by Equations (1) and (2). Writing out Equation (6) results in

$$L_{b,tot} = \left[\varepsilon_p L_b\left(T_p\right) - \varepsilon_w L_b(T_w) - \left(\varepsilon_p - \varepsilon_w\right)L_b(T_{sur})\right]\tau_{atm}x_p + L_b(T_w),\tag{7}$$

Equation (7) illustrates the importance of the water, plastic and surrounding temperatures. It can be shown that the TIR signal, $L_{b,tot}$, is independent of $x_p$ if all temperatures (plastic, water and surroundings) are equal, or if plastic and water emissivity are the same and plastic and water temperatures are the same. If the plastic surface has the temperature of the surroundings, then $L_{b,tot}$ is independent of $\varepsilon_p$ (radiance coming off the plastic is the sum of $\varepsilon_p L_b(T)$ and $(1 - \varepsilon_p)L_b(T)$) but not of $x_p$. If there is a temperature different, the greater the difference between $\varepsilon_p$ and $\varepsilon_w$, i.e., the smaller $\varepsilon_p$, the stronger $L_{b,tot}$ changes with changing floating plastic concentration, enhanced by increasing temperature differences. This is true for broad-band as well as narrow-band sensors. We expect the TIR signal to increase with the increasing temperature difference between the sea and the air above. Peckham et al. [9] assume the temperature of a fibreglass kayak to be approximately that of the water surface, which reduces the slope in Equation (7) to $(\varepsilon_p - \varepsilon_w)[L_b(T_w) - L_b(T_{sur})]\tau_{atm}$. If we assume the temperature of the plastic litter surface to be similar to that of the air above, which also seems reasonable, Equation (7) reduces to $-\varepsilon_w[L_b(T_w) - L_b(T_{sur})]\tau_{atm}$. In both cases, the slope is positive

(negative) when the sea is cooler (warmer) than the air above and steeper when the temperature difference between the sea and above is greater. In the latter case, if the plastic is close to the temperature of the air above, the slope is steepest. It is possible that the plastic is warmer or cooler than either the air or water surface. Examples are, cooling of the plastic surface by a wind blowing over it, or warming of a hollow and transparent litter item by the greenhouse effect. The latter would only be an issue during daylight hours.

Of course, the temperature differences are only one factor in the sensitivity of the TIR signal to changes in plastic concentration; plastic and water emissivity values are the other (Equation (7)). We found a wide range, from 0.10 for polyethylene (PE) to 0.97 for polypropylene (Table 1). Plastic consumer products are usually made of a composition of different polymers and additives. The emissivity of the sea surface for wavelengths used in thermal remote sensing ranges between 0.96 and 0.99 depending on temperature, salinity, angle of view, and sea surface roughness [12]. The dependence of emissivity on salinity is very small, the emissivity difference for standard ocean salinity minus pure-water ranging between ± 0.005 for LWIR [13]. For ice, emissivity is 0.97 to 0.98 in the LWIR range [8]. The emissivity of snow and ice in nadir (straight downward facing) view ranges between 0.95 and 1.00 for wavelengths 10.5 and 12.5 μm, respectively, for different types of snow and ice [14].

**Table 1.** Emissivity values of different polymers and water.

| Material | Emissivity | Range | Reference |
|---|---|---|---|
| Polyester | 0.75–0.85 | thermal | [15] |
| Polyethylene | 0.10 | thermal | [15] |
| Styrofoam, insulating | 0.60 | thermal | [16] |
| Polyvinyl chloride (PVC) | 0.91–0.93 | thermal | [16] |
| Water | 0.98 | 8–14 μm | [8] |
| Ice | 0.97–0.98 | 8–14 μm | [8] |
| Ice | 0.95–0.99 | 10.5–12.5 μm | [14] |
| Snow | 0.97–1.00 | 10.5–12.5 μm | [14] |
| Sea surface | 0.96–0.99 | thermal remote sensing | [12] |

We predict the percentage sensitivity of $L_{b,tot}$ to the presence of floating plastic by defining

$$sens = \frac{dL_{b,tot}}{dx_p} \frac{1}{L_{b,w}} 100\% \tag{8}$$

and calculating *sens* using $dL_{b,tot}/dx_p = L_{b,p} - L_{b,w}$ (Equation (6)) and Equation (5) for $L_{b,w}$ and $L_{b,p}$. In our example calculations (Section 3) we use the spectral band of FLIR (forward-looking infrared) cameras, from 7.5 to 13.5 μm, to derive $L_b$ (Equation (2)). For $\tau_{atm}$ we used 0.97 ± 0.02, the value Minkina and Klecha [17] propose for a distance of 100 m between object and camera, and $\varepsilon_w$ of 0.97 ± 0.03 (Table 1) for the emissivity of the water surface in the FLIR camera range.

### 2.2. Global Temperatures of Seawater and the Temperature Above

It has become apparent in the previous section that the air and sea temperatures are of major importance for the interpretation of TIR remote sensing of the ocean. As these can vary hugely over different locations and during different times of the day/year, this needs further study. We retrieved freely available data from the European Centre for Medium-Range Weather Forecasts (ECMWF) from their "ERA5 hourly estimates of variables on single levels" record [18]. ERA5 is the follow-up for ECMWF's ERA-Interim reanalysis of the global climate. The reanalysis uses the laws of physics to combine model data with satellite and in situ observations from across the world. We used hourly data on a 0.25° × 0.25° grid of sea surface temperature (SST) and temperature at 2 m height (*t2m*) to estimate air–sea temperature differences. For 2007 and later years, ERA5 assimilates data from

OSTIA (Operational Sea Surface Temperature and Sea Ice Analysis) to retrieve SST [19]. OSTIA provides foundation SST at 4–10 m depth where diurnal effects are absent [20]. First, we interpret the results obtained by Topouzelis et al. [5] who took LWIR images at 100 m height using a FLIR Duo R camera of floating targets made of polyethylene terephthalate (PET-1), 1.5 l water bottles, low-density polyethylene (LDPE) plastic bags, and nylon fishing nets in the Aegean Sea in June 2018. Next, we map monthly averages of global temperature differences, SST–$t2m$, at 00:00, 06:00, 12:00, and 18:00 UTC for the months March, June, September, and December in the year 2018. Using these maps, we discuss a number of locations of interest. When reporting SST, $t2m$ or SST-$t2m$, '±' indicates from minimum to maximum over the four time steps. In our example calculations, we use SST for $T_w$ and $t2m$ for $T_{sur}$ and $T_{atm}$. We assume that at the open ocean $T_{sur}$ equals the temperature of the air 2 m above as there are usually no objects, such as buildings or vegetation, that could emit and reflect TIR onto the water and floating plastic. $T_{atm}$ is probably different from $t2m$, especially when the sensor is in the higher atmosphere, but this is not important for our calculation of $L_{b,tot}$ as its radiance term cancels out (Equation (7)).

## 3. Results

### 3.1. Plastic Targets in the Aegean Sea

We interpolated SST and $t2m$ from ERA5 for 07 June 2018 to the coastal waters of Tasmakia beach in the Aegean Sea, Greece, (26.5142° E, 38.9789° N) and found SST of 21.9 °C for 00:00, 06:00, 12:00 and 18:00 UTC while $t2m$ was 21.8, 23.3, 24.9 and 23.7 °C, respectively. The plastic targets were in the water from 08:00 UTC until 17:30 UTC, so an air temperature of 25 °C (and a temperature difference of −3 °C) during thermal imaging is a high estimate. The FLIR spectral band is from 7.5 to 13.5 μm, and we calculated $L_b(T)$ using these wavelengths (Equation (2)). If we assume the surface temperature of the plastic targets to be similar to 25 °C, then *sens* is 4.8% independent of $\varepsilon_p$ (Equation (8)). If instead, we assume the temperature of the plastic surface to be same as SST, we calculate a *sens* of 4.3% and 0.8% for $\varepsilon_p$ of 0.1 (polyethylene) and 0.8 (polyester), respectively. These sensitivities are low for plastic with high emissivity and at sea temperature, and it is, therefore, not surprising that the thermal images of Topouzelis et al. [5] only show the plastic bottles clearly, and the plastic bags with some uncertainty and the fishing nets not at all. The higher visibility of the PET bottles could be because the polyethylene in PET has a low $\varepsilon_p$ (Table 1) and because the greenhouse effect warmed the air and surface inside. The thermal sensitivity of the FLIR camera was not optimised for the conditions and doing so would likely improve the observations of the other litter items.

### 3.2. Global Maps of Monthly Average Air–Sea Temperature Differences

#### 3.2.1. The Month June

The average air–sea temperature difference in the open Aegean Sea was smaller than near Tasmakia beach (Figure 1). Hence, thermal infrared imaging in June is expected to work better near this coast. Around the globe, we found similar air–sea temperature differences to those found at Tasmakia beach in the Arctic Ocean and elsewhere in the northern hemisphere near coasts, in bays and in lakes (Figure 1). In certain areas of the Great Lakes, surface water densities of plastics are as high as those reported for areas of litter accumulation within oceanic gyres [21]. In the middle of the Great Lake 'Lake Michigan' (87° W, 43° N) we found SST of 11.1 °C and $t2m$ of 13.8 ± 0.3 °C with the highest (lowest) $t2m$ at 00:00 (12:00) UTC, resulting in *sens* levels similar to those predicted for Tasmakia beach (Section 3.1). Farther north, in Lake Superior (87° W, 47.5° N), SST-$t2m$ was more negative, SST and $t2m$ being 1.6 and 5.3 ± 0.3, respectively, and estimated *sens* ranged from 1.2% to 6.6%.

A positive temperature difference, sea warmer than air, was more prevalent around the globe. At these locations, plastic would appear dark in the thermal images. The highest levels appeared in the Southern Ocean (Figure 1). Even though it is very remote, Lacerda et al. [22] found plastic pollution

in surface waters near the Antarctic Peninsula. Around here, in the Palmer Basin (65° W, 65° S), SST/*t2m* was −1.0/−2.9 ± 0.1 °C. Numerous research stations are based on the Antarctic Peninsula and the nearby islands. Farther east and south in the open ocean (0° E, 66° S), we estimated SST/*t2m* of −1.6/−6.3 ± 0.2 °C, resulting in a 4.7 °C temperature difference. Using these temperatures, we calculated a *sens* of −8.2% for any $\varepsilon_p$ if the plastic surface temperature is *t2m*, and of −7.4% (−1.4%) if the plastic surface is SST and $\varepsilon_p$ is 0.8 (0.1). High concentrations of surface plastic concentrate in the subtropical gyre in the North Pacific, known as the Great Pacific Garbage Patch (GPGP) [23]. In its centre (140° W, 35° N), SST was 18.7 °C and *t2m* was 17.8 ± 0.3 °C with the lowest *t2m* at 12:00 UTC (03:00 LT). Hence, the temperature difference was a little over 1 °C, implying that thermal imaging could be more challenging in the GPGP. In some areas in the ocean, the temperature difference between seawater and the air above is near zero (Figure 1), and if the plastic surface has the same temperature, thermal imaging would not likely be viable.

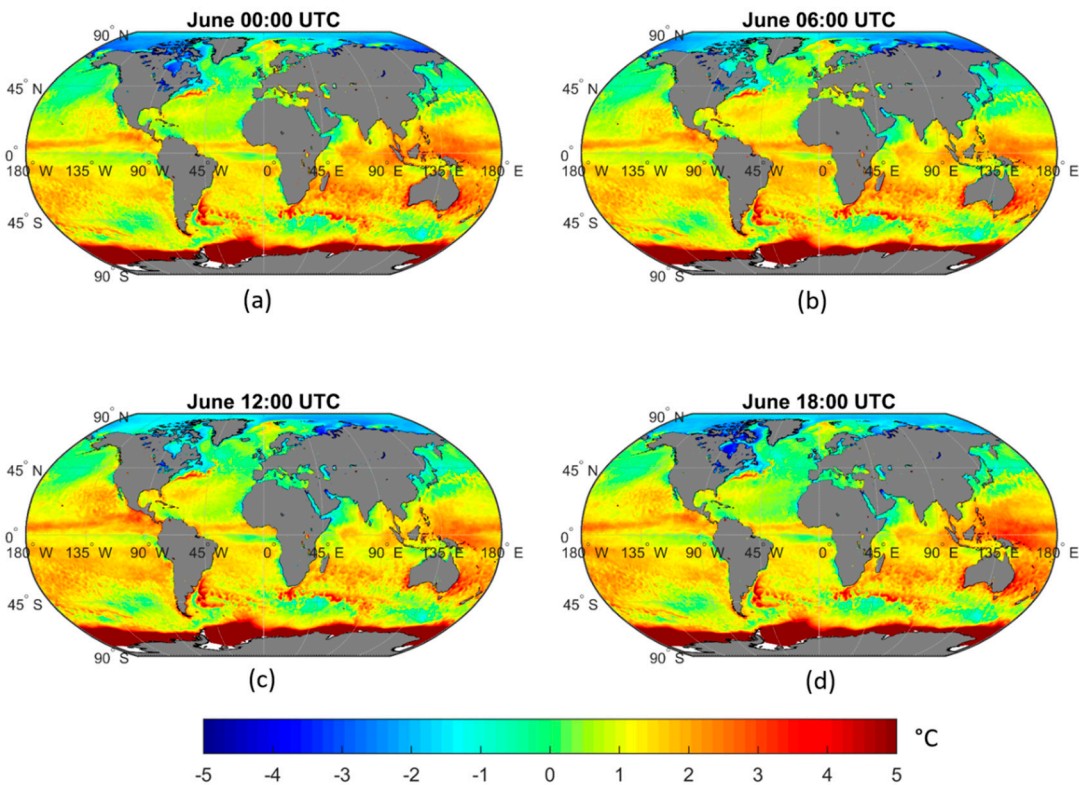

**Figure 1.** Global maps of the monthly average of sea surface temperature–temperature at 2 m height (SST-*t2m*) in the month of June 2018, calculated using ERA5 data [18], for the time steps (**a**) 00:00 UTC; (**b**) 06:00 UTC; (**c**) 12:00 UTC; (**d**) 18:00 UTC.

The global patterns of SST-*t2m* difference did not vary much during the day and night, but locally, for example, in enclosed seas, the variation could be significant due to changing air temperature (Figure 1).

### 3.2.2. The Months December, March and September

The global patterns of SST-*t2m* difference varied substantially over the different seasons, caused by the sun warming the sea and the air above. The water heats and cools slower than the air above because the heat capacity of the oceans is much larger than that of the atmosphere. The maps for the months December, March and September show the resulting transitions (Figures A1–A3). Compared to June, SST-*t2m* in December was reversed in the high and low latitudes with prominent positive

levels in the Arctic Ocean and prominent negative levels in the Southern Ocean (Figure A1). In some areas, such as near the equator, the variation in SST-*t2m* was minor.

At Tasmakia beach in December, SST was 16.4 °C while *t2m* was 10.7 ± 0.7 °C (10.0 °C at 06:00 UTC) and hence, the SST-*t2m* difference was positive, and the absolute difference was double that estimated for June. We calculated *sens* values a little under double (from −1.6% to −8.9%) for those calculated in June, albeit in the negative direction. It may, therefore, be easier to recognise the plastic bags and nets in TIR images in December than in June. Again, the difference was larger near the beach than in the open Aegean Sea. In Lake Michigan in December, we found SST of 4.5 °C and *t2m* of 1.8 ± 0.4 °C, and hence, a positive SST-*t2m* of around 2.7 °C. This resulted in negative calculated *sens* values, similar in magnitude as those found in June. Again, SST-*t2m* was more extreme in Lake Superior. As SST-*t2m* differences look more moderate in March and September (Figures A2 and A3), we expect the TIR signal of floating plastic in the Great Lakes to be weaker than in June or December. In December in the Southern Ocean near the Antarctic Peninsula (65° W, 65° S), SST/*t2m* was −0.8/−0.2 ± 0.3 °C, and therefore, SST-*t2m* was −0.6 °C, a smaller difference than in June. Farther east and south (0° E, −66° N), we estimated SST/*t2m* of −0.6/−1.3 ± 0.2 °C, also a smaller difference than in June. We, therefore, suppose the TIR signal of floating plastic in the Antarctic Ocean to be more pronounced when it is winter in the Southern Hemisphere. Near the GPGP, SST-*t2m* was positive during all seasons (Figures 1 and A1, Figures A2 and A3). At its centre, SST/*t2m* was 17.4/16.4 ± 0.1 °C in December, making the SST-*t2m* difference around 1 °C and similar to June. In September SST-*t2m* was not different, but in March, SST/*t2m* was 15.3/13.0 ± 0.2 °C, raising SST-*t2m* to 2.5 °C at 12:00 UTC. We, therefore, recommend TIR remote sensing of the GPGP in spring and at around 12:00 UTC (04:00 local time) when the air is coldest.

## 4. Discussion

As explained above, TIR remote sensing of floating plastic pollution and how well it works depends on the temperatures of the litter surface, water surface and the atmosphere above. It is expected to be more successful in some places (where the air–sea temperature difference is great) than others. The outcome can be improved by carefully planning the date and time. When surveying from a UAV, the water and air temperature should be taken in concurrence with the TIR measurements to help interpret the recordings. Measuring surface SST is not straightforward. Skin SST, the temperature of the top millimetre of the water and the SST we 'see' with a thermal infrared sensor, is usually lower than that of subskin SST due to a cooling effect of the wind [20]. At lower wind speeds, the vertical temperature profile in the upper metres of the ocean can change during the day due to stratification caused by warming by the sun. Night-time measurements might, therefore, be easier to interpret. Another advantage of night-time measurements is that sun glint would not be an issue for MWIR observations. When using satellite-derived SST, we should not use those based on infrared sensors (e.g., Advanced Very High Resolution Radiometer (AVHRR) and Along Track Scanning Radiometer (ATSR)) but those based on passive microwave sensors instead (e.g., Aqua AMSR-E (Advanced Microwave Scanning Radiometer for Earth Observing System)) [24] (Table 1) to acquire independent temperature data. ERA5 assimilates both thermal infrared and passive microwave as well as in situ SST data. Its SST measurements are, therefore, not completely independent of TIR, but for the theoretical exercise in this paper, we felt this to be acceptable.

Plastic is not the only material that litters natural waters, for example, driftwood and steel drums can float on the surface as well. Polished metals have a low emissivity at 8 to14 μm (0.02–0.2) while for wood, it is 0.87 [8]. It may be possible to separate the TIR signal of different materials, and even different kinds of polymers, by measuring their different spectral signatures, such as with a hyperspectral imager in the NIR-SWIR [4,6]. When applying this methodology, it is still important to aim for the strongest TIR signal as its spectral features weaken when overall levels drop. Emissivity is not only a property of the material, and properties, such as shape and viewing angle, can change the recorded thermal radiance. Another complication is that darker colours absorb more sunlight, get warmer and

emit more thermal radiance than the same material but in clear or light colours. These aspects need further study.

We did not discuss ice coverage in this paper. As the emissivity of snow and ice is comparable to that of liquid water (Table 1), the same theory should apply for plastic littering the frozen water surface. ERA5 does provide parameters such as 'sea ice fraction' and 'temperature of snow layer'.

TIR remote sensing of floating plastic litter could be possible from satellites. The Ocean and Land Colour Instrument (OLCI) with 21 bands in the VIS-NIR on Sentinel-3 works in synergy with Sentinel-3's Sea and Land Surface Temperature Radiometer (SLSTR) comprising six bands in the VIS-SWIR and three in the TIR Ambient bands, 200 K −320 K (3.7–12.0 µm). Its spatial resolution is low, however, 500 m (1 km) for bands in the VIS-NIR (TIR). The Hyperspectral Infrared Imager (HyspIRI) planned by NASA will have a much higher resolution, 30 m for VIS-SWIR and 60 m for TIR (4–13 µm). Although these satellite missions are not specifically designed for marine plastic litter detection, current and future observations such as these could be used.

## 5. Conclusions

We have shown the potential of TIR remote sensing as a floating plastic monitoring tool and its limitations. Because objects emit TIR as well as reflect TIR from their surroundings, the relation between observed thermal radiance and the presence of plastic and emissivity values of water and plastic is not straightforward. We have to take the temperatures of the water and plastic litter surfaces and their surroundings into account. We used a broad-band LWIR camera as an example, but results should be applicable to narrow-band and MWIR sensors as well. TIR sensing is expected to work best at locations and during periods when the water and air temperatures are most different, such as near the South Pole/North Pole in summer/winter. When planning UAV surveys, results can be improved by flying during dates and times of the greatest air–sea temperature differences. When analysing satellite data, the concurrent water and air temperatures need to be known, preferably from non-thermal infrared sensors (e.g., in situ and passive microwave sensors). Night-time observations may be best, as, during daylight hours, the sun can create hot and cold patches on the water surface and produce sun glint (which affects MWIR sensing). TIR remote sensing has not yet been used extensively for monitoring floating plastic, and further measurements are needed to evaluate its usefulness.

**Author Contributions:** Conceptualisation, methodology, formal analysis, investigation, and writing—original draft preparation, L.G.-M.; writing—review and editing, L.G.-M and B.W.

**Funding:** This research received no external funding.

**Acknowledgments:** We acknowledge the use of Copernicus Climate Change Service (C3S) (2017): ERA5: Fifth generation of ECMWF atmospheric reanalyses of the global climate. Copernicus Climate Change Service Climate Data Store (CDS), 30-07-2019. Available from https://cds.climate.copernicus.eu/cdsapp#!/home.

**Conflicts of Interest:** The authors declare no conflict of interest.

## Appendix A

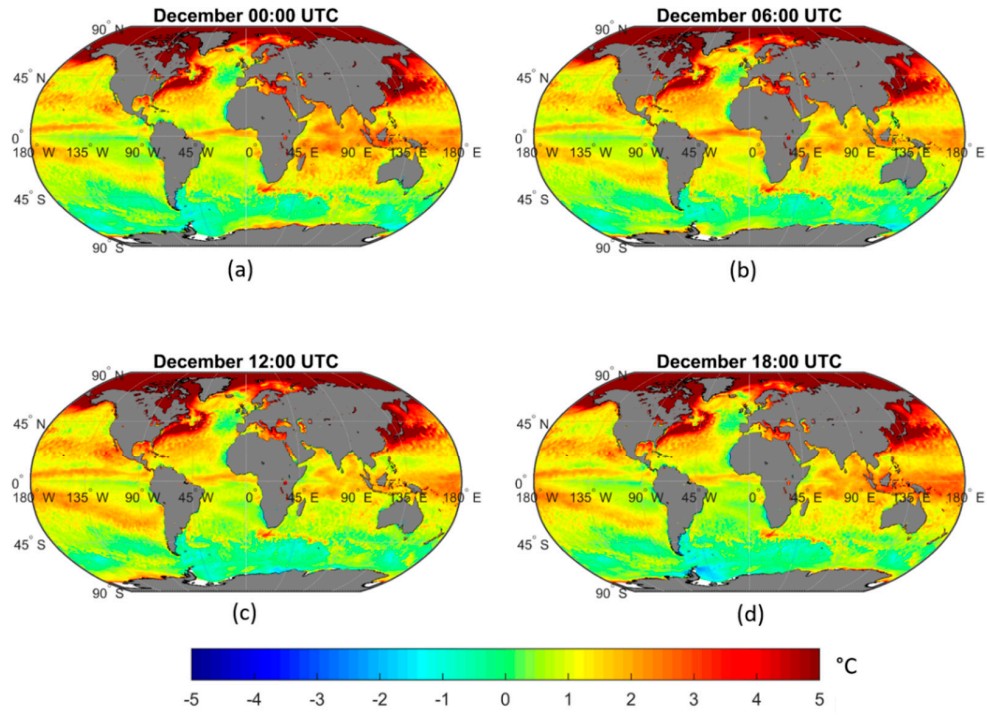

**Figure A1.** Global maps of monthly average of *SST-t2m* in the month of December 2018, calculated using ERA5 data [17], for the time steps (**a**) 00:00 UTC; (**b**) 06:00 UTC; (**c**) 12:00 UTC; (**d**) 18:00 UTC.

## Appendix B

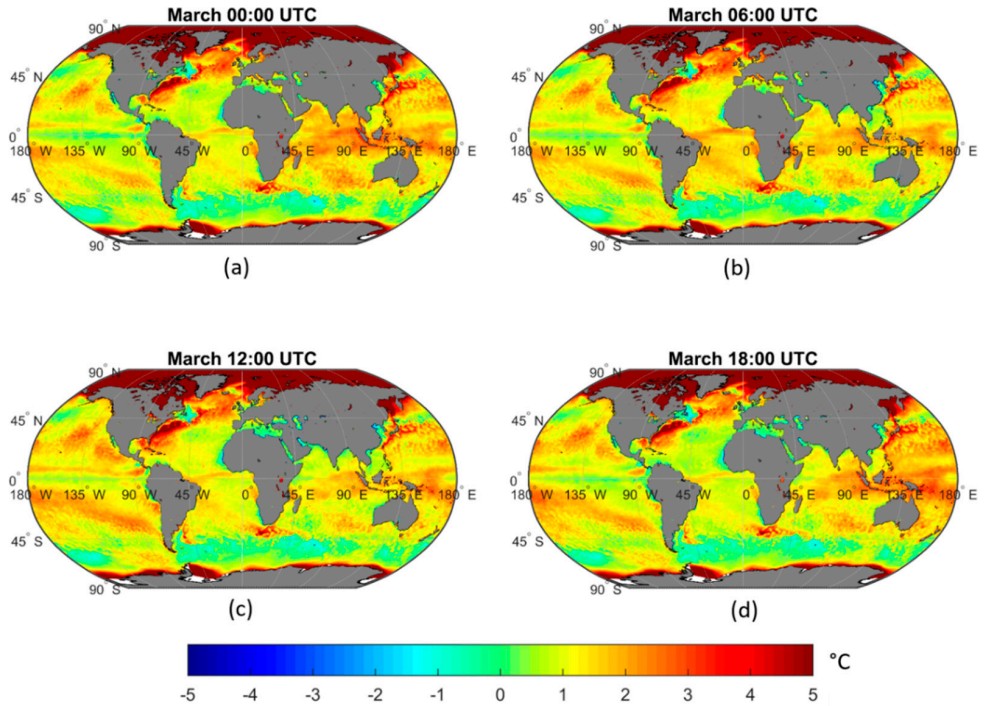

**Figure A2.** Global maps of monthly average of SST-*t2m* in the month of March 2018, calculated using ERA5 data [17], for the time steps (**a**) 00:00 UTC; (**b**) 06:00 UTC; (**c**) 12:00 UTC; (**d**) 18:00 UTC.

**Appendix C**

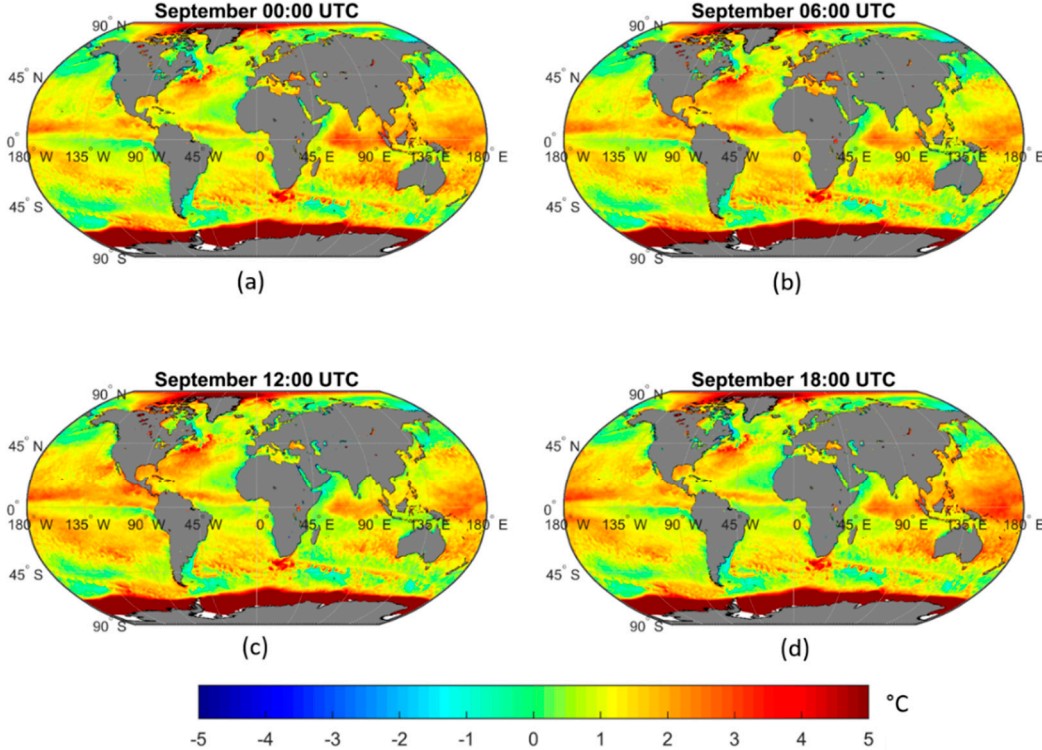

**Figure A3.** Global maps of monthly average of SST-*t2m* in the month of September 2018, calculated using ERA5 data [18], for the time steps (**a**) 00:00 UTC; (**b**) 06:00 UTC; (**c**) 12:00 UTC; (**d**) 18:00 UTC.

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
