# Peer review of "On Thermal Infrared Remote Sensing of Plastic Pollution in Natural Waters"

_remotesensing, doi:10.3390/rs11182159_

Round 1

Reviewer 1 Report

This manuscript intends to use thermal infrared remote sensing to monitoring of plastic pollution of seawater waters. The authors applied radiative transfer theory to predict TIR sensitivity to changes in the surface fraction of water covered by plastic litter and found that the temperature difference between the water surface and the surroundings controls the TIR signal. Generally, the manuscript is well structured, the presentation of the results and discussion is understandable. I have one question about using this method for freshwater and big lakes, is it also possible? If yes, what will be the difference with sea water.

Author Response

We thank the reviewer for their positive feedback and their question. The answer is, yes, this method can be used for freshwater and big lakes. That is why we use the Great Lakes as one of our examples. The dependence of emissivity on salinity is very small. Newman et al. (2005) found measured emissivity difference for LWIR (standard ocean salinity minus pure-water) ranging between ±0.005. We add a reference to Newman et al. (2005) at line 134 of this manuscript.

Newman, S.M.; Smith, J.A.; Glew, M.D.; Rogers, S.M.; Taylor, J.P. Temperature and salinity dependence of sea surface emissivity in the thermal infrared. Q. J. R. Meteorol. Soc. 2005, 131, 2539–2557 doi: 10.1256/qj.04.150

Reviewer 2 Report

The manuscript is proposing an option for detecting plastic pollution in open water bodies, based upon the thermal infrared spectrum.  The authors applied a radiative transfer model taking into account the difference between air temperature at 2 m and the sea surface temperature.  

To the best of my knowledge, the letter presents a pioneering approach.  Different from other approaches relay on the reflectivity bands, thus worth published as a letter.

My main concern, however, is the applied spatial resolution for the floating plastic material.  I guess that in most cases only part of a pixel will be covered.  The authors should assess and discuss what is the area of a single pixel, partially covered by the plastic, that might be detected.  Is a mixture model applicative in this case?

Author Response

We thank the reviewer for their positive feedback and their question. The fraction of the area emitting and reflecting TIR covered by plastic is expressed by xp (line 110). We add “captured by the sensor” to “fraction of the surface” in line 105 to explain this better. The signal is a mixture of that of plastic and that of water (represented by 1- xp) so it is a mixture model. It is difficult to predict the smallest xp that could be detected, it depends on the conditions and the noise of the TIR sensor. This is beyond the scope of this letter.